# Theophylline Induces Remyelination and Functional Recovery in a Mouse Model of Peripheral Neuropathy

**DOI:** 10.3390/biomedicines10061418

**Published:** 2022-06-15

**Authors:** Mert Duman, Stephanie Jaggi, Lukas Simon Enz, Claire Jacob, Nicole Schaeren-Wiemers

**Affiliations:** 1Department of Biology, University of Fribourg, 1700 Fribourg, Switzerland; mert.duman@dbmr.unibe.ch; 2Faculty of Biology, Institute of Developmental Biology and Neurobiology, Johannes Gutenberg University Mainz, 55128 Mainz, Germany; 3Department of Biomedicine, University Hospital Basel, 4031 Basel, Switzerland; stephanie.jaggi@unibas.ch (S.J.); lukas.enz@unibas.ch (L.S.E.); nicole.schaeren-wiemers@unibas.ch (N.S.-W.); 4Department of Biomedicine, University of Basel, 4058 Basel, Switzerland

**Keywords:** hereditary peripheral neuropathy, Charcot-Marie-Tooth disease 1A, demyelination, remyelination, motor function recovery, Theophylline, histone deacetylase 2, Schwann cells, large-caliber axons

## Abstract

Charcot-Marie-Tooth disease (CMT) is a large group of inherited peripheral neuropathies that are primarily due to demyelination and/or axonal degeneration. CMT type 1A (CMT1A), which is caused by the duplication of the *peripheral myelin protein 22* (*PMP22*) gene, is a demyelinating and the most frequent CMT subtype. Hypermyelination, demyelination, and secondary loss of large-caliber axons are hallmarks of CMT1A, and there is currently no cure and no efficient treatment to alleviate the symptoms of the disease. We previously showed that histone deacetylases 1 and 2 (*HDAC1/2*) are critical for Schwann cell developmental myelination and remyelination after a sciatic nerve crush lesion. We also demonstrated that a short-term treatment with Theophylline, which is a potent activator of HDAC2, enhances remyelination and functional recovery after a sciatic nerve crush lesion in mice. In the present study, we tested whether Theophylline treatment could also lead to (re)myelination in a PMP22-overexpressing mouse line (C22) modeling CMT1A. Indeed, we show here that a short-term treatment with Theophylline in C22 mice increases the percentage of myelinated large-caliber axons and the expression of the major peripheral myelin protein P0 and induces functional recovery. This pilot study suggests that Theophylline treatment could be beneficial to promote myelination and thereby prevent axonal degeneration and enhance functional recovery in CMT1A patients.

## 1. Introduction

Charcot-Marie-Tooth disease (CMT) is the most frequent hereditary neuropathy affecting motor and sensory nerves with an estimated prevalence of one in 2500 individuals [1,2]. More than half of all CMT cases are classified as CMT type 1A (CMT1A), characteristic for primary demyelination at an early onset of life [3,4]. The predominant cause for developing CMT1A is overexpression of peripheral myelin protein 22 (PMP22) [5]. Consequently, Schwann cell maturation is disturbed and myelination altered. The peripheral myelin gradually deteriorates, causing demyelination and axonal degeneration, muscle atrophy and weakness, as well as impaired sensation in the extremities [6,7].

To date, there is neither cure nor treatment available that impedes disease progression, and disease management is often a combination of physiotherapy and supportive care [8,9]. Treatment strategies focus on decreasing the expression of PMP22 to recover homeostasis [9,10]. Our approach is different: instead of attempting to decrease the expression of PMP22, we aim at inducing remyelination in Schwann cells to prevent axonal degeneration and to recover function.

The peripheral nervous system (PNS) has high regeneration potential, and upon injury, Schwann cells have the capacity to stimulate axonal regrowth and remyelinate nerve fibers [11,12,13]. We have shown that histone deacetylase 2 (HDAC2) is a critical transcriptional regulator of myelination and remyelination that controls the expression of the myelin transcription factor sex determining region Y-box 10 (Sox10) and of Sox10 target genes including genes coding for pou domain class 3 transcription factor 1 (Pou3f1, aka Oct6) and early growth response-2 (Egr2, aka Krox20) and for myelin proteins such as myelin protein zero (MPZ, aka P0) [14,15,16,17,18,19,20]. Sox10 holds a key role in the entire Schwann cell development and is, therefore, a major transcription factor for Schwann cell differentiation and myelination [21,22,23,24,25]. The transcription factor Oct6 is an intermediate inducer of Schwann cell differentiation, which together with Sox10 activates the transcription of *Krox20* and thereby regulates the timing of myelination [26,27,28]. Krox20 is expressed by myelinating Schwann cells and controls the activation of several myelin genes; Krox20 is thus critical for myelination and for the maintenance of peripheral myelin integrity [28,29,30,31]. These three key transcription factors of myelination coordinate their action in a feed-forward loop: *Sox10* binds to and activates *Oct6* and *Krox20* genes as well as myelin genes, Oct6 synergizes with Sox10 to activate *Krox20* gene, and Krox20 synergizes with Sox10 to activate myelin genes such as *P0*, thereby inducing the program of myelination in the PNS [28,29,30,31,32,33,34,35]. The myelin protein P0 is an integral membrane protein that is almost exclusively expressed in Schwann cells. P0 is the most abundant myelin protein of the PNS. It makes up to 44% of the total myelin proteins and is directly involved in myelin compaction [36,37].

We previously showed that increasing the activity and expression of HDAC2 by a short-term treatment with Theophylline enhances remyelination after a sciatic nerve crush lesion or a spinal cord demyelinating lesion in mice [14]. We showed that when acetylated, eukaryotic elongation factor 1A1 (eEF1A1) translocates to the nucleus where it interacts with Sox10 to drag it out of the nucleus and target it to the proteasome for degradation. In turn, HDAC2 deacetylates eEF1A1, which prevents the relocalization of Sox10 to cytoplasm and its degradation. Increased HDAC2 activity and expression by Theophylline treatment allows maintaining Sox10 on its target genes, thereby increasing the efficiency of remyelination of the PNS and central nervous system (CNS) [14]. Theophylline thus appears to be a potential drug to test in future translational research to accelerate and enhance remyelination following traumatic lesions or in the context of demyelinating diseases.

Theophylline is a methylxanthine derivative that is used as a smooth muscle relaxant, bronchial dilatation agent, diuretic, and cardiac and CNS stimulant. Theophylline is a naturally occurring substance present at very low levels in tea leaves and cocoa beans. It has been widely used as an antiasthmatic and chronic obstructive pulmonary disease drug at high doses, where it acts as an inhibitor of phosphodiesterases, increasing intracellular cAMP levels and protein kinase A activity. At high doses, Theophylline also antagonizes adenosine receptors [38]. Theophylline is also a potent activator of HDAC2, and low doses are sufficient to activate HDAC2 [38]; thus, its therapeutic use as HDAC2 activator would present the advantage of strongly reduced risk of potential adverse events as compared to its use in the treatment of chronic lung diseases. 

In this study, we used a PMP22-overexpressing mouse model (C22) as a model of inherited peripheral neuropathy such as CMT1A [39,40,41]. The C22 mouse genome has been modified using a yeast artificial chromosome construct carrying the human *PMP22* gene. The C22 strain contains seven copies of the human *PMP22* gene. PMP22 overexpression results in a peripheral neuropathy closely resembling the human pathology observed in CMT1A [39]. C22 mice show progressive weakness particularly of the hind limbs and an unstable gait [40]. The microscopic phenotype shows impaired myelinogenesis with widespread hypomyelination and a few hypermyelinated nerve fibers. In particular, medium to large-caliber fibers show reduced myelin sheath thickness, and there is an almost complete absence of myelination of large-caliber fibers with diameter above 5 μm [41].

Here, we show that a short-term low dose Theophylline treatment of C22 transgenic mice induces myelination of large-caliber fibers and leads to improved motor functions. Hence, Theophylline should be further evaluated as a potential new therapeutic approach to sustainably treat demyelinating peripheral neuropathies such as CMT1A.

## 2. Materials and Methods

### 2.1. Animals and Theophylline Treatment

The study included 10 C22 transgenic [41] and 9 control littermate male and female mice. The animals were kept under standard specific pathogen-free conditions and handled according to the guidelines for care and use of experimental animals of the Veterinary Offices of the cantons of Basel (license 1759) and Fribourg (license H-09.2018-FR), Switzerland. Genotypes were determined by PCR using genomic DNA derived from toe clipping biopsies.

Theophylline treatments, functional assays and tissue collection were conducted in the Department of Biology at the University of Fribourg, Switzerland, and the entire procedure was approved by the Veterinary Office of the canton of Fribourg. C22 transgenic and control mice (body weight~15 g) were treated for four consecutive days with either 10 mg/kg Theophylline (Sigma, Buchs, Switzerland) or its vehicle (saline) between 24 and 29 days of age. Mice were randomly allocated to each treatment group with the same number of mice of each sex in each group as much as possible in order to reduce potential sex-related differences.

### 2.2. Inverted Grid Test

Motor functions were tested 21 days after the start of treatment with Theophylline or vehicle using the inverted grid test [42,43] to detect potential functional improvements in motor coordination. Mice were first trained 3 times to the inverted grid test, and afterwards, their performances were recorded 3 times and averaged. 

Mice were placed on a cage grid that was inverted and observed for a fixed duration of 30 s. If the animal fell from the inverted grid, the test was stopped. For each animal, the average counts of grabs (successful steps) and missed steps, as defined by Tilerson et al. [42,43], among the three trials were calculated, as well as the average time they stayed “on” (or attached to) the inverted grid. All trials were video-recorded to allow accurate analysis (Appendix A).

### 2.3. Transmission Electron Microscopy

Mice were sacrificed with a lethal injection of pentobarbital (Esconarkon; Streuli Pharma AG, Uznach, Switzerland). Sciatic nerves were then fixed in situ in 4% paraformaldehyde and 3% glutaraldehyde in 0.1 M phosphate buffer (pH 7.4), all reagents from Sigma-Aldrich (Buchs, Switzerland). Fixed tissues were post-fixed in 2% osmium tetroxide, dehydrated in a graded acetone series, and embedded in Spurr’s resin as previously described [44]. 

Ultrathin sections of 70 nm thickness were cut. Images were acquired using a transmission electron microscope (Philips CM 100 BIOTWIN, Horgen, Switzerland) equipped with a Morada side-mounted digital camera (Olympus, Wallisellen, Switzerland). No contrast was applied. From each ultrathin section, four to five randomly chosen areas of the sciatic nerve were selected and imaged using a 4200× magnification and a resolution of 14.4 nm/pixel.

Per mouse, 40 to 110 single nerve fibers were analyzed. Axons of Remak bundles were not included in the analysis. The G-ratio was calculated by dividing the axonal diameter by the diameter of the whole nerve fiber (axon + myelin sheath). Accordingly, the diameters were derived from the calculated areas and the g-ratio computed. Nerve fibers with no visible myelin sheath were classified as nerve fibers with no compact myelin present. G-ratios of 0.50–0.77 were considered as normally myelinated nerve fibers, values < 0.50 as hypermyelinated nerve fibers and values > 0.77 as hypomyelinated. The counts and measurements were performed using Fiji ImageJ software with the bio-formats importer plugin, version 1.53.

### 2.4. Protein Expression

The fresh frozen sciatic nerves collected for Western blot were homogenized with the Polytron^®^ PT2100 in radioimmunoprecipitation assay (RIPA) buffer (10 mM tris HCl, 150 mM NaCl, 0.5% sodium deoxycholate, 0.1% SDS, 0.5% Nonidet P-40, pH 7.5) containing protease and phosphatase inhibitor mix. The homogenized samples were incubated on ice for 10 min, pelleted, and the supernatant collected. The protein concentration was determined using the Micro BCA Protein Assay Kit (Thermo Fisher Scientific, Zürich, Switzerland). The protein lysates were separated on a 12% sodium dodecyl sulphate polyacrylamide gel electrophoresis (SDS-PAGE) apparatus for 16 h with consistent 18 V to avoid blurry bands from the high lipid content of the samples. The proteins were transferred onto a 0.2-μm polyvinylidene difluoride (PVDF) membrane at 350 mA for 1 h on ice in transfer buffer containing no SDS or methanol. Primary antibodies P0 1:1000 (AvesLab, Davis, CA, USA) and GAPDH 1:5000 (Genetex #GTX28245, LubioScience GmbH, Zürich, Switzerland) were incubated overnight at 4 °C. The IRDye coupled secondary antibodies 800 CW (#926-32218) and 680 RD (#926-68070) from LI-COR Biosciences (Bad Homburg, Germany) were diluted 1:10,000. Membranes were excited at 700 and 800 nm, captured, and the fluorescent bands quantified using the Odyssey CLx System from LI-COR and Image Studio software, version 5.

### 2.5. mRNA Expression

Total RNA from sciatic nerve tissues was extracted using the lithium chloride/urea RNA precipitation method, as previously described [45]. Each sample was homogenized in 0.5 mL 3 M LiCl/6 M urea using Polytron^®^ PT2100 (Kinematica AG, Malters, Switzerland) and incubated overnight at 0 °C. The pellet was dissolved in 250 μL 10 mM Tris HCl (pH 7.6), 1 mM EDTA and 0.5% EDTA, and an equivalent amount of phenol:chloroform:isoamylalcohole (25:24:1) (Invitrogen #15593-031, Thermo Fisher Scientific, Basel, Switzerland) and incubated for 10 min. The samples were centrifuged, and the RNA in the top aqueous layer precipitated in 50 μL 3 M sodium acetate (pH 5.2), 1 mL ethanol absolute and 1 μg glycogen (Roche #10901393001, Mannheim, Germany) overnight at −20 °C. The extracted RNA was dissolved in 15 μL nuclease-free water and the concentration determined using Nanodrop™ One (Thermo Fisher Scientific, Basel, Switzerland).

Synthesis of cDNA was performed according to the manufacturer’s instructions: GoScript™ Reverse Transcription System (Promega #A2800) with random primers. 

Real-time qPCR reactions were carried out according to the manufacturer’s recommendations: GoTaq^®^ qPCR System (Promega #A6002, Dübendorf, Switzerland). For the amplification, 2 μL 1:5 diluted cDNA was used. The thermal cycling was run with the 7500 fast real-time PCR system (Applied Biosystems, Thermo Fisher Scientific, Zürich, Switzerland). All primers used were designed using the NCBI PrimerBLAST^®^ tool and ordered from MicroSynth (Balgach, Switzerland). The final concentration in master mix was 0.5 μM. Primer sequences were as follows: *P0* forward, 5′-GCCCGCTAACCGCTATTTC-3′; *P0* reverse, 5′-CCTCTCAGGTCACGCTCTATG-3′; *Krox20* forward, 5′-CAGGGTACTGTGGGTCAATGG-3′; *Krox20* reverse, 5′-TCTACCCGGTGGAAGACCTC-3′. For the reference genes, primer pairs for beta-actin (Actb), ribosomal subunit 60S (60S), hypoxanthine-guanine phosphoribosyltransferase (Hprt), 2 different pairs for the ribosomal subunit 40S (40S-1 and 40S-2), ornithine decarboxylase antizyme 1 (Oaz1), and ribosomal protein S13 (Rps13) were tested and assessed by the geNorm algorithm for stability [46]. Actb, 60S, and Hprt were chosen for normalization of sciatic nerve tissue of C22 and control mice due to highest stability. Primer sequences were as follows: *Actb* forward, 5′-ACCCGCGAGCACAGCTTCTT-3′; *Actb* reverse, 5′-ACATGCCGGAGCCGTTGTCG-3′); *60S* forward, 5′-GGAAGTACCAGGCAGTGACAG-3′, *60S* reverse, 5′-GCAGGCATGAGGCAAACAG-3′; *Hprt* forward, 5′-TCAGACGGCTTTTTGCCGCGA-3′; *Hprt* reverse, 5′-TCTCGA GCAAGTCTTTCAGTCCTGT-3′.

The raw qPCR data from the 7500 Fast Real-Time PCR System software (version 2.3) was exported into an Excel file. The gene expression levels were quantified relative to the reference gene expression without efficiency correction [47].

### 2.6. Statistical Analysis

*p*-values were calculated in Microsoft Excel (Mac version 16.34) using a one-tailed or two-tailed Student’s *t*-test. A *p*-value < 0.05 was considered significant. Individual data points are represented in the graphs by scatter points. Data are presented as mean values, error bars are SEM. No animal or data point was excluded from the analysis. 

## 3. Results

### 3.1. Theophylline Treatment Leads to Partial Functional Recovery in C22 Transgenic Mice

Three- to four-week old C22 transgenic and control mice were treated for four consecutive days with Theophylline or its vehicle. Twenty-one days after the start of treatment, motor coordination performance was tested using an inverted grid test. Mice were sacrificed 2 days later for tissue collection of sciatic nerve pairs. One sciatic nerve was fixed and embedded for electron microscopy, while the other sciatic nerve was divided into two pieces, one for Western blot and one for quantitative reverse transcription PCR (qRT-PCR) analysis (Figure 1A).

The three trials of grab- and miss-counts, as well as the time mice remained on the inverted grid (attached to the grid without falling into the cage placed underneath) were averaged. Theophylline-treated C22 transgenic mice showed more successful grabs of the grid and fewer missed steps as compared to vehicle-treated C22 transgenic mice (Figure 1B). The average time C22 mice stayed on the grid (Figure 1C) and the total number of steps (Figure 1D) were similar for Theophylline- and vehicle-treated mice, suggesting that improved performance (decreased percentage of missed steps) of C22 mice treated with Theophylline was not due to a change in muscular strength or a general stimulating effect of the treatment. In comparison, control littermate mice did not miss any step (Figure 1B) and remained on the inverted grid the full duration of the test (Figure 1C). Interestingly, the total number of steps was significantly higher in control mice treated with Theophylline as compared to control mice treated with vehicle (Figure 1D), suggesting a general stimulation effect of the Theophylline treatment, which was, however, not observed in C22 transgenic mice.

### 3.2. Theophylline Treatment Induces (Re)Myelination in C22 Transgenic Mice

To test whether the 4-day Theophylline treatment induced (re)myelination in C22 transgenic mice, we collected sciatic nerves 2 days after the inverted grid test for electron microscopy, Western blot analysis and qRT-PCR. 

By electron microscopy, we analyzed the percentage of myelinated axons and myelin thickness. For myelin thickness, we calculated the g-ratio, which was plotted as a function of the axon diameter to detect potential correlations (Figure 2A–D). As previously described [39,40,41], we observed a deficiency of myelinated large-caliber fibers > 4 μm in C22 transgenic mice as compared to control mice, and the g-ratio scatter diagram of C22 transgenic mice showed that the degree of myelination decreased with increasing axon diameter (Figure 2A–D). The frequency distribution of the axon diameter of myelinated axons was similar in C22 transgenic mice treated with Theophylline or with vehicle, except for the thinly wrapped large-caliber axons > 4 μm appearing in the scatter diagram of the *C22* transgenic mice treated with Theophylline (Figure 2A,B). These observations warranted closer examination of the large-caliber axons > 4 μm. The average percentage of counted large-caliber axons > 4 μm was comparable in C22 transgenic mice treated with Theophylline or with vehicle (Figure 2E). However, the nerves of C22 transgenic mice treated with Theophylline showed that a majority of these large-caliber axons were wrapped by a thin myelin sheath (Figure 2F, filled arrowheads, and Figure 2G). In comparison, the majority of these large-caliber axons were unmyelinated in C22 transgenic mice treated with vehicle (Figure 2F, empty arrowheads, and Figure 2G). In addition, we found a significant difference in the percentage of myelinated large-caliber axons > 4 µm between nerves of Theophylline-treated and of vehicle-treated C22 transgenic mice (Figure 2G), but there was no difference in the average g-ratio of all myelinated axons (Figure 2H), indicating that the treatment with Theophylline induced (re)myelination of large-caliber axons but did not affect myelin thickness of already myelinated axons.

As previously shown [14], Theophylline treatment had no effect on the g-ratio (Figure 2H) or on P0 protein expression levels (Figure 3A) in control mice, and we did not find any unmyelinated axons in control mice, except for Remak bundle axons (Figure 2F,G). However, consistent with the induction of remyelination by Theophylline treatment in C22 transgenic mice, P0 protein levels were significantly increased in C22 transgenic mice treated with Theophylline as compared to C22 transgenic mice treated with vehicle (Figure 3A). At the mRNA level, there was a trend for increased *P0* and *Krox20* in C22 transgenic mice treated with Theophylline as compared to C22 mice treated with vehicle, whereas no trend was detected in control mice treated with Theophylline as compared to vehicle (Figure 3B).

## 4. Discussion

Efficacious therapy of the hereditary primary demyelinating neuropathy CMT1A is required to promote functional recovery of the peripheral nerves and improve patients’ outcome. Low-dose short-term Theophylline treatment was shown to enhance myelination after a sciatic nerve crush injury via activation of the *Sox10* transcriptional pathway [14]. In the present pilot study, the effects of a short-term Theophylline treatment in C22 transgenic mice were investigated on functional, ultrastructural, as well as protein and mRNA levels.

Functional testing of C22 transgenic mice showed significant improvement of motor coordination, suggesting that Theophylline treatment activated the myelination process in these mice, thereby leading to partial functional recovery. However, the average time Theophylline-treated C22 transgenic mice stayed attached to the grid was the same as for vehicle-treated C22 transgenic mice, likely indicating no difference in muscle strength. A possible explanation could be that building up muscles takes a longer time and is strongly dependent on exercise, which may be restricted within the cage. This point was not further assessed. 

Increased myelination after Theophylline treatment was confirmed by transmission electron microscopy analyses of C22 transgenic sciatic nerves. Indeed, while most large-caliber axons in C22 transgenic mice treated with vehicle showed no signs of myelination, a majority of large-caliber axons of C22 transgenic mice treated with Theophylline were surrounded by a thin myelin sheath. Our observations led to the question of why increased myelination was predominantly observed in large-caliber axons. A study conducted by Robaglia-Schlupp et al. demonstrated that in C22 transgenic mice, Schwann cells of large-caliber axons remain in an immature non-myelinating stage [48]. In line with this idea, one could speculate that Theophylline initiated myelination in arrested Schwann cells of large-caliber axons via the activation of Sox10 target genes. The diameter of axons strongly correlates with the nerve type, e.g., motor nerves contain a larger amount of large-caliber axons as compared to sensory or mixed nerves [49]. This suggests that Theophylline may not affect all Schwann cells evenly in C22 transgenic mice, but may preferably act on Schwann cells surrounding motor axons. The significantly better scoring of C22 transgenic mice treated with Theophylline as compared to vehicle-treated mice in the inverted grid test supports the hypothesis of improved motor function. However, transmission electron microscopy did not allow us to distinguish the type of nerve fibers by their morphological appearance. Identification of motor and sensory fibers can be performed by immunohistochemical analyses [50], but the limited number of study animals, i.e., amount of sciatic nerve tissue did not allow examination by immunohistochemistry.

Theophylline leads to increased expression of HDAC2. In turn, HDAC2 maintains Sox10 on its target genes by preventing its degradation [14]. We have previously shown that a short Theophylline treatment of 20 h starting at 11 days post sciatic nerve crush lesion in adult mice leads to increased recruitment of Sox10 to its target genes *Krox20* and *P0* and that HDAC2, Sox10, Krox20 and P0 protein levels are increased after a Theophylline treatment of 2 days or 4 days starting at 10 days post sciatic nerve crush lesion [14]. At 30 days post lesion, i.e., 16 days after the end of Theophylline treatment, while Krox20 and P0 protein levels were still higher in sciatic nerves of Theophylline-treated mice as compared to vehicle-treated mice, the levels of HDAC2 and Sox10 were back to baseline [14], suggesting that at this time-point after treatment, the recruitment of Sox10 to its target genes was not promoted anymore. Here, we tested the mRNA levels of Sox10 target genes *Krox20* and *P0* at day 23, i.e., 19 days after the end of Theophylline treatment. Indeed, while P0 protein levels were increased in sciatic nerves of C22 transgenic mice treated with Theophylline as compared to C22 mice treated with vehicle, we did not detect significant changes in *Krox20* or *P0* mRNA levels. However, we found a trend for increased *Krox20* and *P0* mRNA levels in sciatic nerves of C22 transgenic mice treated with Theophylline as compared to C22 transgenic mice treated with vehicle, whereas no trend was observed in sciatic nerves of control mice treated with Theophylline as compared to mice treated with vehicle. Taken together with our previous data showing normalized HDAC2 and Sox10 protein levels 16 days after the end of Theophylline treatment [14], this suggests that tissues should be collected at an earlier time-point after treatment to detect a significant increase in *Krox20* and *P0* mRNA levels, as demonstrated for the Theophylline treatment after sciatic nerve crush lesions [14]. 

Control mice showed no difference in ultrastructural, protein or mRNA levels between Theophylline- and vehicle-treated mice. This is consistent with the findings of Duman et al. [14], who showed that Theophylline did not affect the myelination process of stably myelinating Schwann cells in intact nerves.

This pilot study presents preliminary but significant data showing the relevance of Theophylline treatment in hereditary demyelinating diseases such as CMT1A. The small cohorts served to obtain evidence-based data and to define relevant conditions for a full-scale research project. This study was, however, limited by the small sample size, which made it difficult to examine different treatment approaches and time-points after treatment. A short-term Theophylline treatment of 4 days was sufficient to induce (re)myelination of large-caliber axons and partial functional recovery in C22 transgenic mice. It would be interesting in future studies to test different Theophylline treatment paradigms, such as longer-term treatments or several treatment rounds, to potentially further improve (re)myelination and functional recovery, and to test the efficacy of Theophylline treatment in other models of demyelinating peripheral neuropathies. A recent study showed that treatment with an HDAC3 inhibitor could also improve function and myelination in another CMT1A mouse model [51]. Since HDAC3 inhibition and HDAC2 activation by Theophylline treatment act through different pathways, it would be interesting to test whether a combination treatment could have additive or synergistic effects in the induction of (re)myelination and functional recovery in demyelinating peripheral neuropathies. Another limitation of this study lies in the animal model employed, which mimicked some features of peripheral neuropathies in humans but not all. For example, neuropathic pain could not be studied with our C22 transgenic mice, which mainly displayed a defect in myelination of large-caliber motor axons. Other models need to be used to identify potential therapeutic agents counteracting neuropathic pain [52]. Interestingly, treatment with other HDAC modulators such as MS-275, which is an inhibitor of HDAC1 and HDAC3, has been shown to modulate neuropathic pain in pre-clinical studies [53,54]. It would be interesting to test whether a combination therapy with MS-275 and Theophylline leads to improvement of motor activity and decreases neuropathic pain in the context of demyelinating peripheral neuropathies. 

## 5. Conclusions

In recent years, methylxanthines such as Theophylline have been reported in various studies to have potential beneficial effects to counteract neurodegenerative diseases including Parkinson’s diseases, Alzheimer’s disease and multiple sclerosis [14,55]. In particular, we have shown that Theophylline promotes remyelination in the lysolecithin-induced mouse model of multiple sclerosis lesion, through activating and upregulating HDAC2 [14], warranting future clinical trials to test the efficiency of Theophylline in remyelinating lesions of multiple sclerosis patients. In the present study, our data indicated that Theophylline appears also as a promising treatment option for primary demyelinating disorders such as CMT1A. Sustainably treating demyelination in CMT1A patients or in the context of other demyelinating peripheral neuropathies has the potential to lead to functional recovery, prevent axonal degeneration, and thereby help to significantly improve patients’ quality of life.

## Figures and Tables

**Figure 1 biomedicines-10-01418-f001:**
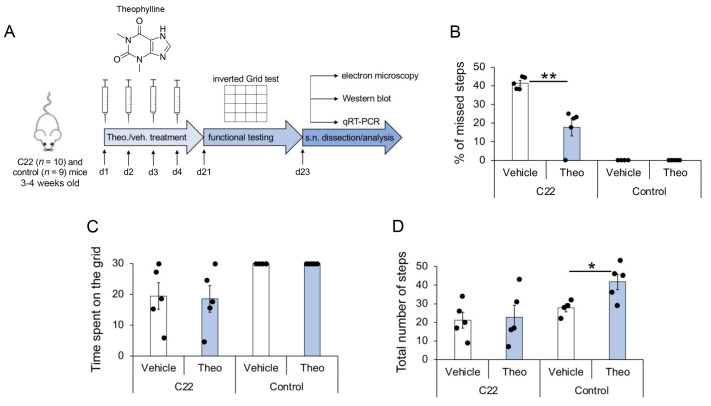
Flowchart of study setup and partial functional recovery of C22 mice by Theophylline treatment. (**A**) C22 and control littermate mice were injected with 10 mg/kg Theophylline (Theo.) or with vehicle (veh.) for four consecutive days (d1–d4), such as described in Ref. [14]. Motor function was analyzed by inverted grid test at d21, and sciatic nerves (s.n.) were dissected at d23 for quantitative analysis of myelination by electron microscopy, Western blot and qRT-PCR. Five C22 and 4–5 control mice per treatment group were used. Graphs showing the percentage of missed steps (**B**), the average time spent on the inverted grid (**C**) and the total number of steps (**D**) observed in C22 and control littermate mice treated with Theophylline (Theo) or with vehicle. Unpaired two-tailed Student’s *t*-test, *n* = 5 (C22 mice) or 4–5 (Control mice) per treatment group, *p* value: * < 0.05, ** < 0.01. Black dots show individual data points. See also Appendix A for statistical values.

**Figure 2 biomedicines-10-01418-f002:**
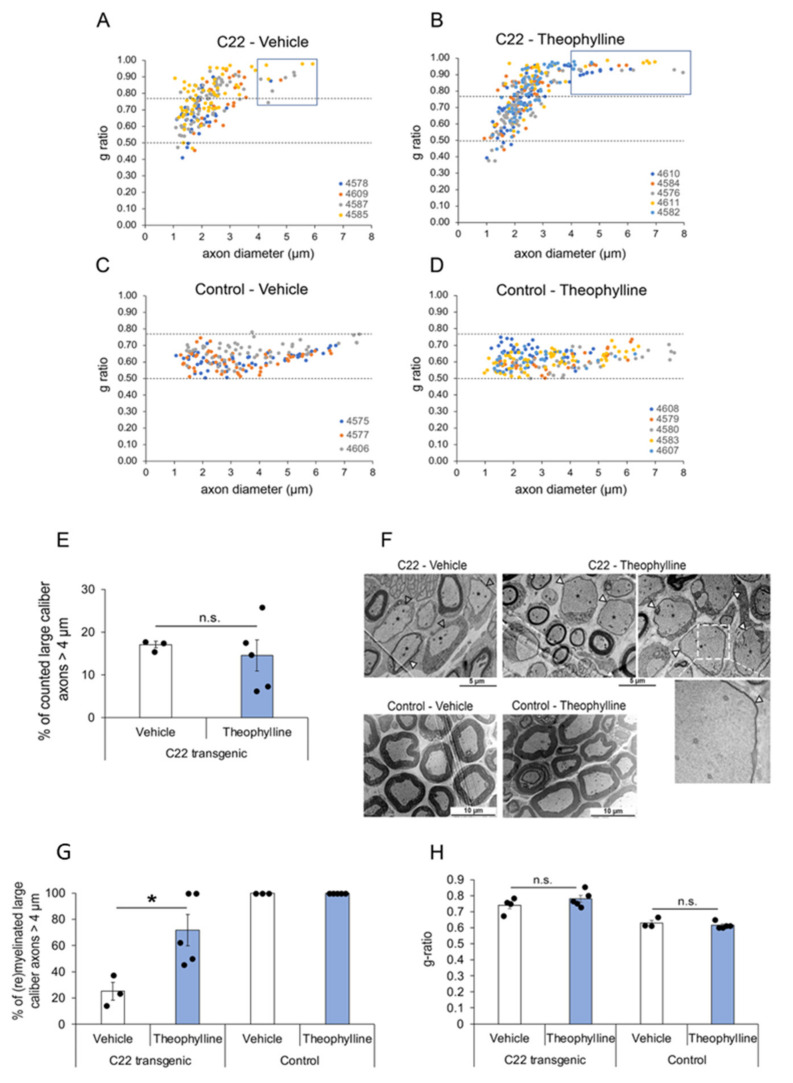
Theophylline induces (re)myelination of large-caliber axons. G-ratio distribution in sciatic nerves of C22 transgenic and control mice treated with Theophylline or vehicle. G-ratio/axon diameter scatter diagrams of sciatic nerve axons from C22 transgenic (**A**,**B**) and control (**C**,**D**) mice treated with Theophylline or vehicle as shown in Figure 1A. The data points of one animal (each identified by a unique number) are in the same color. Values above 0.77 indicate hypomyelinated axons, while values below 0.50 indicate hypermyelinated axons (delineated by dashed lines). For C22 transgenic mice treated with vehicle (**A**), the g-ratio of 203 axons is plotted (*n* = 4 mice, 25–80 axons per mouse). For C22 transgenic mice treated with Theophylline (**B**), the g-ratio of 308 axons is plotted (*n* = 5 mice, 35–79 axons per mouse). For control mice treated with vehicle (**C**), the g-ratio of 178 axons is plotted (*n* = 3 mice, 55–62 axons per mouse). For control mice treated with Theophylline (**D**), the g-ratio of 214 axons is plotted (*n* = 5 mice, 31–65 axons per mouse). The blue boxes in (**A**,**B**) indicate large-caliber axons > 4 µm. (**E**) Graph representing the percentage of counted large-caliber axons (>4 µm diameter) in sciatic nerves of C22 mice treated with vehicle or Theophylline. (**F**) Representative electron micrographs of sciatic nerves of C22 or control mice treated with vehicle or Theophylline. The dashed white box on the top right image highlights the region corresponding to the magnified image underneath showing a thin myelin sheath (filled arrowhead) around this large-caliber axon. (**G**) Graph representing the percentage of (re)myelinated large caliber axons (>4 µm) in sciatic nerves of C22 or control mice treated with vehicle or Theophylline. Unpaired two-tailed Student’s *t*-test, *n* = 3 for vehicle-treated C22 or control mice and 5 for Theophylline-treated C22 or control mice. (**H**) Graph representing the average g-ratio of all counted axons in sciatic nerves of C22 or control mice treated with vehicle or Theophylline. * *p* value < 0.05, n.s.: statistically non-significant. (**E**,**H**) Unpaired one-tailed Student’s *t*-test, *n* = 3 for vehicle-treated C22 or control mice and 5 for Theophylline-treated C22 or control mice. Black dots show individual data points. See Appendix A for statistical values. Treatment with vehicle and Theophylline and time-point of tissue collection were as shown in Figure 1A.

**Figure 3 biomedicines-10-01418-f003:**
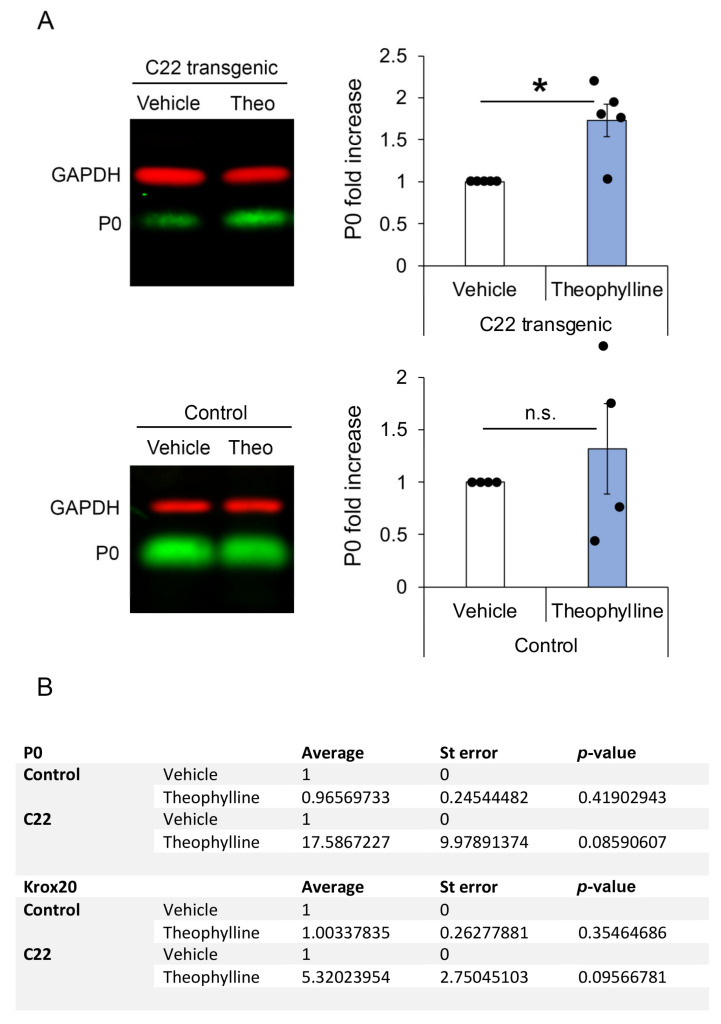
Theophylline treatment leads to increased myelin protein zero (P0) at the protein level. (**A**) Representative Western blot images of P0 and GAPDH (loading control) in sciatic nerve lysates of C22 or control mice treated with vehicle or Theophylline and graphs representing the fold increase of P0 protein levels (normalized to GAPDH) in samples of Theophylline-treated as compared to vehicle-treated mice (vehicle samples set to 1). The fluorescent signal of the IRDyes was detected and quantified using Odyssey CLx Imaging Software. Paired two-tailed (for C22) or one-tailed (for control) Student’s *t*-test, *n* = 5 C22 and 4 control mice per treatment condition (vehicle or Theophylline), *p* value: * < 0.05, n.s.: non-significant. Black dots show individual data points. See Appendix A for statistical values. (**B**) Table showing *P0* and *early growth response-2* (*Krox20*) mRNA levels in sciatic nerves of C22 and control mice treated with vehicle or Theophylline and collected 19 days after treatment. Paired one-tailed Student’s *t*-test, *n* = 5 C22 and 4 control mice per treatment condition. See Appendix A for statistical values.

## Data Availability

All data are available from the corresponding author (C.J.).

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
