# Peer review of "Theophylline Induces Remyelination and Functional Recovery in a Mouse Model of Peripheral Neuropathy"

_biomedicines, 2022, doi:10.3390/biomedicines10061418_

Round 1
Reviewer 1 Report
Paper titled (Theophylline Induces Remyelination and Functional Recovery 2
in a Mouse Model of Peripheral Neuropathy) by Duman et al. is well organized but have some draw backs the most important is lacking the rationality & utility. Why authors thought to use theophylline? it is known as high risk medication and has many untolerable adverse effects !! Can authors recomemnd the drug to patients with NP ( along lasting condition) ?!! I dont think so.
1- Methods: no data about age and weight of the mice
2- No clear experimental design and grouping is demonstrated
3- Is the inverted grid test is valuable for measuring neuropathy (allodynia& hyperalgesia)??
4- Were mice naive to the test or trained on it?
5- The title is not informative and did not tell what was the mechanism by which theophylline produced these improveemnts
6- Discussion is long
7- Introduction did not provdie a rational why authors thought the drug may improve NP
Author Response
Comments and Suggestions for Authors
Paper titled (Theophylline Induces Remyelination and Functional Recovery 2
in a Mouse Model of Peripheral Neuropathy) by Duman et al. is well organized but have some draw backs the most important is lacking the rationality & utility. Why authors thought to use theophylline? it is known as high risk medication and has many untolerable adverse effects !! Can authors recommend the drug to patients with NP (a long lasting condition) ?!! I dont think so.
Answer: Our aim with this study was to test whether a short-term theophylline treatment could induce or promote myelination or remyelination in a model of peripheral neuropathy such as CMT1A. The idea of using theophylline to induce/promote (re)myelination directly comes from our previous study Duman et al., Nat. Commun., 2020, where we confirmed that theophylline is a potent activator of HDAC2 (this was already known, but had not been tested in myelinating cells) when used at low doses, where theophylline does not target phosphodiesterases and adenosine receptors, its other main known targets. We have also shown in this study that a short-term treatment (only 4 days) of theophylline promotes remyelination after a sciatic nerve crush injury and also after a demyelinating lesion in the spinal cord. We have shown that when acetylated, eukaryotic Elongation Factor 1A1 (eEF1A1) translocates into the nucleus of Schwann cells or oligodendrocytes, where it interacts with the transcription factor Sox10 to drag it out of the nucleus and target it to the proteasome for degradation. We have shown that HDAC2 deacetylates nuclear eEF1A1 to send it back to the cytoplasm and maintain Sox10 on its target genes, which are promyelinating factors and myelin proteins. We have shown that theophylline increases the expression of HDAC2, Sox10, Krox20 and P0, decreases the levels of acetylated eEF1A1, and promotes remyelination and functional recovery after a sciatic nerve crush lesion. Based on these findings, we thought that theophylline could also induce or promote myelination or remyelination in the context of a peripheral neuropathy such as CMT1A. So, the rationale of using theophylline in a CMT1A model was to promote myelination. The advantage of using theophylline as a remyelination treatment is that low doses are sufficient to activate HDAC2 and increase its expression, so the risk of adverse effects is a lot decreased as compared to a treatment for asthma or COPD where high doses must be used to inhibit phosphodiesterases. In addition, we have seen that a short-term 4-day treatment with theophylline is sufficient to induce or promote (re)myelination for at least 2 weeks, either after a sciatic nerve crush lesion, a spinal cord demyelinating lesion or in the context of the peripheral neuropathy model that we have tested in this study, so the treatment does not need to be continuously taken and can be taken by rounds of 4 days every two weeks. Of course, more work is needed to determine the optimal treatment regimen in each type of disorder. To our opinion, this would be best tested by clinical studies on specific patient groups. We have extended the introduction to further explain the rationale of the study and its potential utility for the treatment of peripheral neuropathies.
1- Methods: no data about age and weight of the mice
Answer: The information about the age and weight of the mice appears in the Materials and Methods section, under 2.1 Animals and Theophylline treatment, 2nd paragraph: "C22 transgenic and control mice (body weight ~ 15 g ) were treated for four consecutive days with either 10 mg/kg Theophylline or its vehicle (saline) between 24 and 29 days of age."
2- No clear experimental design and grouping is demonstrated
Answer: We are not sure that we understand which information is missing here. Figure 1A of our revised manuscript (Figure 1 in our initial submission) shows the experimental design. Concerning the rationale of this design, we have added more information in the Results section, under 3.1 Theophylline Treatement Leads to Partial Functional Recovery in C22 Transgenic Mice. We have also extended the Introduction section to further explain the rationale of our study. Concerning C22 transgenic and control mice, we have only used littermate mice. This is indicated in the Materials and Methods section, under 2.1. Animals and Theophylline treatment. So, the C22 transgenic mice and the control mice are from the same litters. Then, to group the mice in the two different experimental treatments, Theophylline or vehicle, we have randomly allocated mice to each group with the same number of mice of each sex in each group as much as it was possible, so that we can reduce inter-individual differences. We have made that point clearer in the Materials and Methods section, under 2.1 Theophylline Treatment, 2nd paragraph.
3- Is the inverted grid test valuable for measuring neuropathy (allodynia& hyperalgesia)??
Answer: No, the inverted grid test is not a test to measure sensory function, but motor function. The treatment with Theophylline leads to myelination of large-caliber axons, which are motor axons. The small caliber axons are a lot less affected in C22 mice.
4- Were mice naive to the test or trained on it?
Answer: Mice were first trained 3 times to the inverted grid test and afterwards their performances were recorded 3 times and averaged. We have added this information in the Materials and Methods under 2.2 Inverted Grid Test, 1st paragraph.
5- The title is not informative and did not tell what was the mechanism by which theophylline produced these improveemnts
Answer: The mechanism by which Theophylline induces (re)myelination is not addressed in this study, because it was already extensively studied in Duman et al., Nat. Commun., 2020. The rationale of testing the promyelinating effect of Theophylline in C22 mice is based on the mechanism we published in this previous work. We don’t have an idea of a different title, the current title summarizes the findings of the present study, so we would like to keep this title.
6- Discussion is long
Answer: We are not sure how to answer this comment since Reviewer 3 thinks that the discussion is too short. We are doing our best to write an appropriate, concise and informative discussion and we hope to be able to satisfy our three reviewers.
7- Introduction did not provdie a rational why authors thought the drug may improve NP
Answer: We have developped the introduction to explain better the rationale of this study. We hope that our revised manuscript is more informative.
Reviewer 2 Report
Duman and colleagues investigate theophylline, a phosphodiesterase inhibitor currently used in therapy for asthma and other respiratory diseases, to be a potential remyelination-inducing drug. They previously published and showed that theophylline treatment enhances remyelination and functional recovery after sciatic nerve crush lesion in mice (Nat. Commun. 2020). In the current pilot set of experiments, they provide functional, ultrastructural, as well as molecular data on a Pmp22-overexpressing mouse model (C22) which results in a peripheral neuropathy closely resembling the human pathology observed in hereditary CMT subtype 1A (CMT1A). All in all, this is a good work that strengthens their previous one.
The following comments should be taken into consideration:
Line 99: (Error! Reference source not found.).
In 2.6. Statistical Analysis it is not necessary to elaborate on the 3 excluded occasions. Although they would like to be transparent about the results, I would remove lines 191-198, for the sake of brevity.
Line 202. “with either 10 mg/kg Theophylline or vehicle (such as described in Ref. 6)” It’s better to incorporate this information in fig.1 and then add the reference as well.
Fig 1 can be merged with Fig 2 which include the experimental setup and motor tests. I would also advise to incorporate the molecular structure of theophylline (or at least the chemical formula) to help readers better understand the compound properties.
Fig 3 can be merged with part of Fig 4 (the axonal analyses with g-ratios) as a cohesive, single figure.
Fig 4D Western blot images can be separated in a third figure since it’s a molecular technique and added with Krox20 and P0 mRNA levels mentioned in the last part of the results but have no representation in a figure yet, whatsoever.
Additionally, for this last part of the result section (Line 285-310) Most of the information provided should be moved to the discussion section since it’s based on their previous Nat. Commun publication and do not belong in the current manuscript.
Line 319, motoric -> motor
Author Response
Comments and Suggestions for Authors
Duman and colleagues investigate theophylline, a phosphodiesterase inhibitor currently used in therapy for asthma and other respiratory diseases, to be a potential remyelination-inducing drug. They previously published and showed that theophylline treatment enhances remyelination and functional recovery after sciatic nerve crush lesion in mice (Nat. Commun. 2020). In the current pilot set of experiments, they provide functional, ultrastructural, as well as molecular data on a Pmp22-overexpressing mouse model (C22) which results in a peripheral neuropathy closely resembling the human pathology observed in hereditary CMT subtype 1A (CMT1A). All in all, this is a good work that strengthens their previous one.
Answer: We thank Reviewer 2 for the positive comments on our study
The following comments should be taken into consideration:
Line 99: (Error! Reference source not found.).
Answer: We don’t understand which reference was not found. Figure 1 was just underneath line 99. In any case, we have removed the reference to a figure there since it seems confusing and we have merged initial Figure 1 with initial Figure 2 in a single figure. So, initial Figure 1 is now Figure 1A in our revised manuscript.
In 2.6. Statistical Analysis it is not necessary to elaborate on the 3 excluded occasions. Although they would like to be transparent about the results, I would remove lines 191-198, for the sake of brevity.
Answer: We have removed the explanation on the 3 exclusions, as requested.
Line 202. “with either 10 mg/kg Theophylline or vehicle (such as described in Ref. 6)” It’s better to incorporate this information in fig.1 and then add the reference as well.
Answer: We have added this information in Figure 1’s legend (now, Figure 1A) together with the reference.
Fig 1 can be merged with Fig 2 which include the experimental setup and motor tests. I would also advise to incorporate the molecular structure of theophylline (or at least the chemical formula) to help readers better understand the compound properties.
Answer: We have merged initial Figures 1 and 2 into a single figure (now Figure 1), and we have added the chemical formula of Theophylline to Figure 1 (now, Figure 1A)
Fig 3 can be merged with part of Fig 4 (the axonal analyses with g-ratios) as a cohesive, single figure.
Answer: We have merged these two figures into a single figure (now, Figure 2)
Fig 4D Western blot images can be separated in a third figure since it’s a molecular technique and added with Krox20 and P0 mRNA levels mentioned in the last part of the results but have no representation in a figure yet, whatsoever.
Answer: We have separated the WB data of P0 and we have added the mRNA data in a tabular form. Here, we don’t think that we should put a lot of emphasis on the mRNA data because, as we explained in the manuscript, the time-point of tissue collection is too late to catch a significant regulation at the mRNA level.
Additionally, for this last part of the result section (Line 285-310) Most of the information provided should be moved to the discussion section since it’s based on their previous Nat. Commun publication and do not belong in the current manuscript.
Answer: As requested, we have moved that part into the Discussion section
Line 319, motoric -> motor
Answer: We have corrected that.
Reviewer 3 Report
3 May 2022
Regarding the review of manuscript ‘Theophylline Induces Remyelination and Functional Recovery in a Mouse Model of Peripheral Neuropathy’ by Jacob C et al., submitted to Biomedicines
Manuscript ID: biomedicines-1696215
Dear Authors,
Charcot-Marie-Tooth disease (CMT) is a group of hereditary disorders that cause motor and sensory neuropathy. Meanwhile, theophylline has been shown to enhance remyelination and functional recovery after a sciatic nerve crush lesion in mice. In the present review entitled ‘Theophylline Induces Remyelination and Functional Recovery in a Mouse Model of Peripheral Neuropathy’, Jacob and colleagues investigated the effects of theophylline on myelination, peripheral myelin protein P0, and nerve functions in the peripheral myelin protein 22-overexpressed mouse model of CMT. The main strength of this original review article is that it addresses an interesting and timely question, revealing a short-term low dose theophylline increases the percentage of myelinated large-caliber axons, the expression of the major peripheral myelin protein P0, and induces functional recovery.
In general, I think the idea of this article is really interesting and the authors’ fascinating observations on this timely topic may be of interest to the readers of Biomedicines. However, some comments, as well as some crucial evidence that should be included to support the author’s argumentation, needed to be addressed to improve the quality of the manuscript, its adequacy, and its readability before its publication in the present form. My overall judgment is to publish this article after the authors has carefully considered my suggestions below, reshaping parts of the abstract and the body of manuscript by reorganizing the introduction to orderly present concepts in general, rationales, and the purpose of this study. In general, I recommend the authors to use more evidence to back their claims, especially in the in introduction, the discussion, and the conclusion, which I believe are currently insufficient. Thus, I advise the authors to attempt to deepen the subject of their manuscript and to focus their efforts on researching relevant literature: I believe that adding more studies will help to provide better and more accurate background to this paper. In this study, I will try to help the author by presenting comments below to improve their manuscript.
Please consider the following comments:
- Abstract: Please expand the abstract with up to 200 words, presenting the background, the rationale and purpose, the methods, the results, and the conclusion proportionally.
- Keywords: I recommend listing up to ten keywords.
- A graphical abstract is highly recommended.
- Introduction: The section is well presented, but too concise. I recommend presenting ample background regarding theophylline, including its effects on injured nerves and its mechanisms. It also deserves to describe the rationales and potential significance to study the effects of theophylline in CMT, ending with the purpose of this study.
- Results: Please present statistical values in tabular form.
- Discussion: In my opinion, this research article would be more compelling and useful to a broad readership if the authors moved beyond and discussed theoretical and methodological avenues in need of refinement, using this evidence to suggest a path forward. The section also benefits from describing the potent, weakness, limitation of this study, technology necessary to achieve this goal, and future research direction.
- References: The number of references too low for original research like this. The original article like this manuscript needs at least 60-70 references. I strongly recommend searching relevant studies in literature, which support the expertise of the authors and expand the introduction and discussion sections particularly. According to the Journal’s guidelines, the authors should revise the bibliography for all the references. Also, they should have provided the DOI number for each reference. Suggested references: https://doi.org/10.3390/ijms222011055; https://doi.org/10.3390/biomedicines10010018; https://doi.org/10.3390/ijms23010098; https://doi.org/10.3390/biomedicines9080897.
Overall, the manuscript contains four figures, no table and 32 references. I believe that the manuscript may carry important value the beneficial effects of theophylline on nerves in animal model of peripheral neuropathy. I hope that, after these careful revisions, the manuscript can meet the Journal’s high standards for publication. I am available for a new round of revision of this review.
I declare no conflict of interest regarding this manuscript.
Best regards,
Reviewer
Author Response
Comments and Suggestions for Authors
3 May 2022
Regarding the review of manuscript ‘Theophylline Induces Remyelination and Functional Recovery in a Mouse Model of Peripheral Neuropathy’ by Jacob C et al., submitted to Biomedicines
Manuscript ID: biomedicines-1696215
Dear Authors,
Charcot-Marie-Tooth disease (CMT) is a group of hereditary disorders that cause motor and sensory neuropathy. Meanwhile, theophylline has been shown to enhance remyelination and functional recovery after a sciatic nerve crush lesion in mice. In the present review entitled ‘Theophylline Induces Remyelination and Functional Recovery in a Mouse Model of Peripheral Neuropathy’, Jacob and colleagues investigated the effects of theophylline on myelination, peripheral myelin protein P0, and nerve functions in the peripheral myelin protein 22-overexpressed mouse model of CMT. The main strength of this original review article is that it addresses an interesting and timely question, revealing a short-term low dose theophylline increases the percentage of myelinated large-caliber axons, the expression of the major peripheral myelin protein P0, and induces functional recovery.
In general, I think the idea of this article is really interesting and the authors’ fascinating observations on this timely topic may be of interest to the readers of Biomedicines. However, some comments, as well as some crucial evidence that should be included to support the author’s argumentation, needed to be addressed to improve the quality of the manuscript, its adequacy, and its readability before its publication in the present form. My overall judgment is to publish this article after the authors has carefully considered my suggestions below, reshaping parts of the abstract and the body of manuscript by reorganizing the introduction to orderly present concepts in general, rationales, and the purpose of this study. In general, I recommend the authors to use more evidence to back their claims, especially in the in introduction, the discussion, and the conclusion, which I believe are currently insufficient. Thus, I advise the authors to attempt to deepen the subject of their manuscript and to focus their efforts on researching relevant literature: I believe that adding more studies will help to provide better and more accurate background to this paper. In this study, I will try to help the author by presenting comments below to improve their manuscript.
Answer: We thank Reviewer 3 for the positive comments and suggestions for improvements of our manuscript.
Please consider the following comments:
- Abstract: Please expand the abstract with up to 200 words, presenting the background, the rationale and purpose, the methods, the results, and the conclusion proportionally.
Answer: We have extended the abstract to around 200 words to give more details about the background, rationale and purpose.
- Keywords: I recommend listing up to ten keywords.
Answer: We have listed ten keywords
- A graphical abstract is highly recommended.
Answer: We have added a graphical abstract
- Introduction: The section is well presented, but too concise. I recommend presenting ample background regarding theophylline, including its effects on injured nerves and its mechanisms. It also deserves to describe the rationales and potential significance to study the effects of theophylline in CMT, ending with the purpose of this study.
Answer: We have extended the introduction as requested
- Results: Please present statistical values in tabular form.
Answer: We have added an excel file as supplementary tables presenting statistical values
- Discussion: In my opinion, this research article would be more compelling and useful to a broad readership if the authors moved beyond and discussed theoretical and methodological avenues in need of refinement, using this evidence to suggest a path forward. The section also benefits from describing the potent, weakness, limitation of this study, technology necessary to achieve this goal, and future research direction.
Answer: We have modified the discussion as requested.
- References: The number of references too low for original research like this. The original article like this manuscript needs at least 60-70 references. I strongly recommend searching relevant studies in literature, which support the expertise of the authors and expand the introduction and discussion sections particularly. According to the Journal’s guidelines, the authors should revise the bibliography for all the references. Also, they should have provided the DOI number for each reference. Suggested references: https://doi.org/10.3390/ijms222011055; https://doi.org/10.3390/biomedicines10010018; https://doi.org/13390/ijms23010098; https://doi.org/10.3390/biomedicines9080897.
Answer: We tried to find other relevant references for our study, but we did not find many. All the references suggested by Reviewer 3 are about sensory neuropathy/pain, but our study is on recovery of motor function and remyelination of large-caliber axons (motor axons). Therefore, we do not know where to include the suggested references. However, one of the references suggested by Reviewer 3 is on the potential beneficial effect of an HDAC inhibitor MS-275 in neuropathic pain. So, we have added a paragraph in the discussion explaining that our C22 mice do not model neuropathic pain and that other mouse models should be used to identify potential therapies to treat neuropathic pain. We have added a reference to a review on mouse models of neuropathic pain and as an example of therapy, we have added the reference suggested by Reviewer 3 on the use of MS-275 in neuropathic pain and another reference on the same topic. We also added the DOI number of all references. We submitted our manuscript as a short communication because it is a pilot study, this is why it does not have an extensive list of references. We hope that the list of references is acceptable now.
Overall, the manuscript contains four figures, no table and 32 references. I believe that the manuscript may carry important value the beneficial effects of theophylline on nerves in animal model of peripheral neuropathy. I hope that, after these careful revisions, the manuscript can meet the Journal’s high standards for publication. I am available for a new round of revision of this review.
Round 2
Reviewer 1 Report
I find the paper is still not suitable for publication in biomedicines
Author Response
I find the paper is still not suitable for publication in biomedicines
Answer: We are sorry that Reviewer 1 still does not think that our manuscript fits to Biomedicines. We have however addressed all comments of Reviewer 1 and now since there are no additional comment or request to improve our manuscript, we do not know what is still not acceptable for Reviewer 1.
Reviewer 3 Report
27 May 2022
Regarding the review of manuscript ‘Theophylline Induces Remyelination and Functional Recovery in a Mouse Model of Peripheral Neuropathy’ by Jacob C et al., submitted to Biomedicines
Manuscript ID: biomedicines-1696215
Dear Authors,
The authors did an excellent work clarifying the questions I have raised in the previous round of review. Currently, this paper is a well-written and timely piece of research and provides a useful study investigating the effects of theophylline on myelination, peripheral myelin protein P0, and nerve functions in the peripheral myelin protein 22-overexpressed mouse model of Charcot-Marie-Tooth disease.
I presented some minor comments below which may help improve the quality of this manuscript and still suggest citing more references to back up the arguments presented by the authors with evidence.
Comments:
1. Abstract: Please expand the abbreviation CMT1A for the first appearance such as CMT type 1A (CMT1A).
2. Introduction: Please expand the abbreviations CMT and HDAC2 for the first appearance in the body of manuscript.
3. Results: The figures should be presented in color.
4. References: The number of references too low for original research like this. The original article like this manuscript needs at least 50 references. I strongly recommend searching relevant studies in literature, which support the expertise of the authors and expand the introduction and discussion sections particularly.
Overall, this is a timely and needed work, thus I believe that manuscript now meets the Journal’s standards for publication. I am always available for other reviews of such interesting and important articles. I look forward to seeing further study on this issue by these authors in the future.
Thank you for your work.
I declare no conflict of interest regarding this manuscript.
Best regards,
Reviewer
Author Response
The authors did an excellent work clarifying the questions I have raised in the previous round of review. Currently, this paper is a well-written and timely piece of research and provides a useful study investigating the effects of theophylline on myelination, peripheral myelin protein P0, and nerve functions in the peripheral myelin protein 22-overexpressed mouse model of Charcot-Marie-Tooth disease.
I presented some minor comments below which may help improve the quality of this manuscript and still suggest citing more references to back up the arguments presented by the authors with evidence.
Answer: We thank Reviewer 3 very much for appreciating our revisions and also for suggesting additional improvements. We have worked on the remaining comments. Here are below our answers.
Comments:
- Abstract: Please expand the abbreviation CMT1A for the first appearance such as CMT type 1A (CMT1A).
Answer: We have expanded the abbreviation as requested.
- Introduction: Please expand the abbreviations CMT and HDAC2 for the first appearance in the body of manuscript.
Answer: We have done so.
- Results: The figures should be presented in color.
Answer: We have modified the figures to present them in color.
- References: The number of references too low for original research like this. The original article like this manuscript needs at least 50 references. I strongly recommend searching relevant studies in literature, which support the expertise of the authors and expand the introduction and discussion sections particularly.
Answer: We have included additional references as requested. Our manuscript cites now 55 references.
Overall, this is a timely and needed work, thus I believe that manuscript now meets the Journal’s standards for publication. I am always available for other reviews of such interesting and important articles. I look forward to seeing further study on this issue by these authors in the future.
Thank you for your work.
I declare no conflict of interest regarding this manuscript.